# NISF++: Geometrically-grounded implicit representations of 3D+time cardiac function from 2D short- and long-axis MR views

**Nil Stolt-Ansó** [1,2] iD                                                                 NIL.STOLT@TUM.DE
**Maik Dannecker** [2,3] iD                                                        MAIK.DANNECKER@TUM.DE
**Steven Jia** [4] iD                                                                  STEVEN.JIA@UNIV-AMU.FR
**Julian McGinnis** [1,2,3] iD                                                     JULIAN.MCGINNIS@TUM.DE
**Daniel Rueckert** [1,2,3,5] iD                                                  DANIEL.RUECKERT@TUM.DE

[1] *Munich Center for Machine Learning, Technical University Munich, Germany*

[2] *School of Computation, Information and Technology, Technical University Munich, Germany*

[3] *TUM University Hospital, Technical University Munich, Germany*

[4] *Institut de Neurosciences de la Timone, Aix-Marseille Université, France*

[5] *Department of Computing, Imperial College London, United Kingdom*

**Editors:** Accepted for publication at MIDL 2026

## Abstract

Clinical acquisition in cardiac magnetic resonance (CMR) imaging involves obtaining cross-sectional planes of the heart along the radial and longitudinal directions. Despite these planes being 2D cross-sectional images of the heart, radiologists understand the 3D spatial and continuous temporal nature of the organ being imaged. The same can not be said about the conventional deep learning architectures used to process CMR images, which rely on in-plane and grid-based operations, and are hence unable to organically integrate information from all imaging planes. In this paper, we build upon previous work on neural implicit segmentation functions (NISF) to overcome unaddressed challenges in cardiac function modeling in the CMR domain.

For a given subject, our architecture builds a shared 3D+time representations from all available acquisition planes regardless of orientation. By design, predictions along any imaging plane orientation are cross-sections of the same 3D representation, leading to spatio-temporal consistency across all slices. Moreover, our architecture makes the rotation and translation parameters of imaging planes learnable, allowing us to correct for the commonplace respiratory and patient motion between slice acquisitions under a rigid assumption. Furthermore, interpolation of intensities and segmentation can be performed in 4D at any desired resolution. We perform our study on a 120 subject sub-cohort of CMR imaging data from the UK-Biobank. We show our in-plane segmentation performance to be on-par with existing CMR segmentation methods and explore how the majority of failure cases arise from limitations in the ground-truth segmentation, for which our representations make predictions with better anatomical accuracy than its original training data. We also evaluate our motion-correction capabilities, displaying quantitative and qualitative improvements in slice alignment. Our qualitative results explore how our representations can be derived irrespective of missing acquisition planes and opens up avenues towards modeling complex sub-structures such as papillary muscles.

**Keywords:** cardiac magnetic resonance imaging, neural implicit functions, segmentation, motion correction.

## 1. Introduction

When analyzing medical images, radiologists possess an excellent understanding of the continuous 3D structure and function of the organs being imaged. This anatomical prior in human perception recognizes magnetic resonance images are cross-sectional slices of 3D volumes and each voxel is a discrete intensity samples of an otherwise continuous space. This intuition informs us on how an imaged tissue interpolates between pixels/voxels and how the content of the images extrapolates beyond the image boundaries.

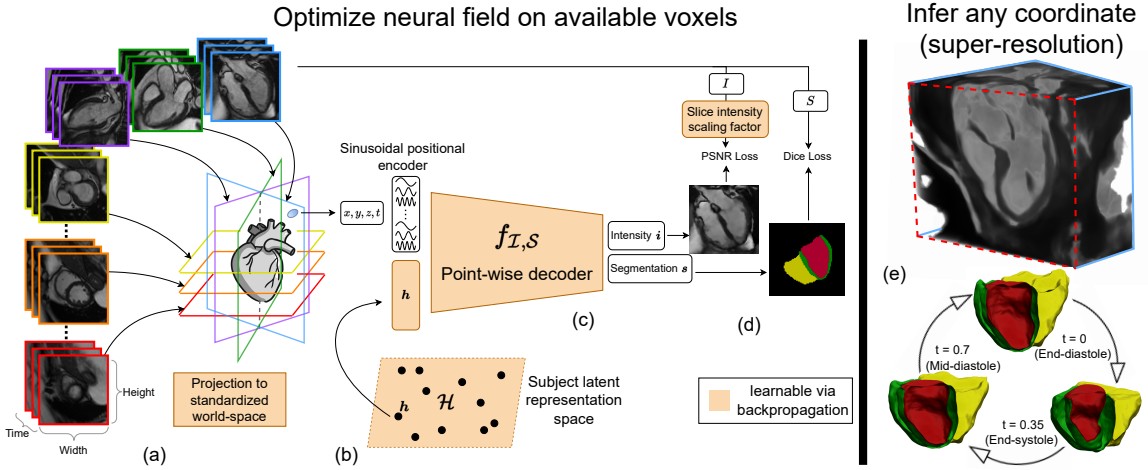

Figure 1: Overview of the proposed framework: **a)** Images are transformed to a standardized world-space. The NISF model performs rigid-body motion correction to account for patient and/or respiratory motion. **b)** Voxel coordinates are encoded via sinusoidal features and concatenated with (learnable) subject representation vector. **c)** The decoder predicts intensity and segmentation values for each input voxel. **d)** Intensity and segmentation predictions are supervised. Supervision includes physics-informed constraints specific to magnetic resonance imaging. **e)** At inference, any coordinate can be queried, allowing for super-resolution of intensity and segmentation fields beyond the imaging planes.

On the other hand, the current paradigm of deep learning (DL) architectures for image-processing, such as U-Net (Ronneberger et al., 2015) or Vision Transformer (Dosovitskiy, 2020) variants, are explicitly built around grid-based operations and possess no such inductive bias over the content of these images. These architectures have no awareness beyond the input grid of intensities presented to them, simply learning to extract statistical patterns in pixel correlations and mapping these to task-dependent outputs.

This difference is especially evident in the context of modeling cardiac morphology and function. In CMR imaging, the range of imaging plane orientations, while providing radiologists insights across all necessary regions of the heart, makes it impossible to organically package all image information for standard grid-based DL architectures. Moreover, these

networks can only produce in-plane predictions, which require hand-crafted post-processing heuristics to extract a 3D or 4D model. By performing slice-wise processing of CMR images, it is not possible to exploit correlations between different views of the cardiac anatomy. Moreover, the overwhelming majority of works process images 'as-is', disregarding any form of inter-slice motion due to respiratory or patient motion that is commonplace in real-world data.

### 1.1. Motivation

In this work, we introduce an architecture aiming to organically handle the nuances of CMR data and integrating inductive biases more inline with those exhibited by radiologists. Our approach achieves this by employing neural implicit functions (NIF), a family of point-wise decoders that model representations of continuous fields (Park et al., 2019; Mildenhall et al., 2021). This study extends existing works in the CMR domain (Stolt-Ansó et al., 2023; Banus et al., 2025) by handling unaddressed challenges in CMR image-processing. We do so by incorporating any number of arbitrarily-oriented views, being able to correct for inter-slice motion, performing spatio-temporally consistent segmentation along all views, and introducing physics-informed constraints specific to CMR to perform super-resolution. Our contributions can be summarized as follows:

- We propose a novel architecture based for neural implicit functions that is capable of modeling 3D+time representations of cardiac function by aggregating all available 2D+time views regardless of orientation. These representations can be leveraged to perform segmentation and super-resolution tasks.

- We showcase our approach's ability to correct inter-slice misalignments due to patient or respiratory motion directly from image information based on its implicit understanding of a subject's anatomy.

- We demonstrate our approach's ability to generalize beyond the presented imaging planes by making predictions along held-out views.

- We introduce various physics-informed constraints to more accurately regularize the learning of intensity fields in CMR imaging data.

We make the code repository publicly available and provide a project page with further interactive visualizations of our results[1].

## 2. Background

In the clinical setting, cardiaovascular function is assessed by quantifying volumetric and deformation metrics of the left ventricle. To calculate these metrics, standard CMR imaging protocols acquire slices in multiple orientations. Short-axis (SA) slices are a series of imagine planes descending along the left ventricle used to asses radial myocardial motion and provide volumetric measurements. Since the SA stack offers very limited resolution along the

---

1. Project page: CMR_NISF.github.io

ventricles' longitudinal direction, longitudinal motion is assessed via perpendicular imaging planes at various angles termed long-axis (LA) views.

Early works (Bai et al., 2018) showed that the extraction of these metrics could be automated by processing each individual CMR slice using a deep learning segmentation architecture. Subsequent works have attempted to incorporate all available views (Chen et al., 2019; Zhang et al., 2024) by aggregating information in the latent space. However, these works are only able to produce predictions in-plane and the segmentations are not guaranteed to be mutually-consistent across views. Recent works propose architectures capable of out-of-plane predictions (Stolt-Ansó et al., 2023; Banus et al., 2025), but exclusively apply them to the SA stack.

One aspect of CMR image processing that is predominantly ignored in deep learning works is slice-misalignments due to respiratory and subject motion. A wide range of works attempt to perform motion correction based on explicit statistical shapes (Mitchell et al., 2003; Meng et al., 2023; Beetz et al., 2022). The generation process of these explicit atlases and their generalization to rare morphologies remains a significant challenge. Furthermore, these approaches typically base the motion correction solely on shape information and do not directly include image intensities into the inference process. Alternative approaches attempt to correct for motion without shape information using exclusively image intensities based on slice intersections (Stolt-Ansó et al., 2024) or using an additional 3D volume as reference (Chandler et al., 2008). While parallel lines of research outside of cardiac imaging have explored joint optimization methods for motion estimation and super-resolution (Xu et al., 2023), CMR imaging remains an open challenge due to the sparse number of slices acquired in clinical CMR protocols.

## 3. Method

### 3.1. Data overview

We create a small scale sub-cohort of 120 subject from the UK-Biobank (Sudlow et al., 2015) with a 100-10-10 subject split for training, validation, and testing phases. While our method can be scaled to larger dataset sizes, we limit the scope of this work to a small cohort in order to show the efficacy of our method. Each subject contains 2-, 3-, and 4-chamber LA CMR cine acquisitions, as well as a SA cine stack composed of 8-13 slices per subject. Since each SA slice is an individual acquisition (like each LA slice), the SA stack is split into its constituting slices. Each 2D+time slice has their intensities clipped based on the upper 99th percentile and then normalized within a $[0, 1]$ range. Finally, a segmentation network (Bai et al., 2018) is employed to segment the left ventricle (LV) and right ventricle (RV) on SA slices, resulting in 4 distinct classes: background, LV blood pool, LV myocardium, and RV blood pool. In the absence of automatic LA segmentations, we perform manual segmentation for 3 distinct time points along the cine sequence for all 10 testing subjects and 20 of the 100 training subjects.

### 3.2. Architecture

**Standardized world coordinates.** The set of CMR slices belonging to a given subject carry DICOM header information regarding how these slices intersect in world-space. Our

goal is to treat the collection of images of a subject as a point cloud of voxels in world space that describe the intensity and segmentation values present at a given coordinate.

To allow our representations to exploit anatomical similarities across subjects, our pre-processing pipeline automatically creates a standardized coordinate system where every subject's anatomy is roughly aligned to a canonical pose. Despite pre-alignment, deviations in anatomy may still be present due to respiratory and patient motion, as well as variability in acquisition plane positions. We address the correction of this motion later in this section.

By employing DICOM header information, we re-orient the collection of slices of a given subject such that the normal of the SA slices aligns to the coordinate system's Z-axis. Next, we orient the mid-ventricular LV-RV line to be parallel to the Z-axis by using the intersection line between 4-chamber LA plane and the mid-ventricular SA plane. We find the center of the LV myocardium between left and right ventricles along the aforementioned LV-RV line, and designate this as the origin of the coordinate system. Finally, we crop images around the new coordinate origin within a $(80, 80)$ pixel region and normalize a subject's scanner coordinates projection to result within a $[-1, 1]$ range.

**Network architecture.** Our architecture is composed of a multi-layer perceptron with weights $\psi$ to learn a mapping $f_\psi$ from a spatiotemporal coordinate system $\Omega \subset \mathbb{R}^4$ to a $d$-dimensional signal $f_\psi : \Omega \to \mathbb{R}^d$. The network's input follows a DeepSDF architecture (Park et al., 2019), being composed of a 3D coordinate $\mathbf{x}$, a time coordinate $t$, and a subject-specific learnable conditioning vector $\mathbf{h} \in \mathbb{R}^{128}$. We define the full 3D+time coordinate as $\mathbf{c} = (x, y, z, t) \in \Omega$. We find that a ReLU-based architecture with a sinusoidal positional encoder similar to be more robust, memory efficient, and easier to parameter-tune, while producing indistinguishable modeling fidelity to other alternatives.

The network $f_\psi$ learns to model two continuous scalar fields: intensity $\mathcal{I} : \Omega \to [0, 1]$ and segmentation probabilities $\mathcal{S} : \Omega \to [0, 1]^4$ of all four distinct labels. We denote the network as $f_\psi(\mathbf{x}, t, \mathbf{h}) = (f_\mathcal{I}, f_\mathcal{S})$ which is implemented as a shared MLP body with 16 residual layers of 256 rectified linear units with two output heads corresponding to $f_\mathcal{I}$ and $f_\mathcal{S}$ fields. While the same network weights $\psi$ are shared across all subjects, subject-specific characteristics are modeled using a given subject's learnable conditioning vector $\mathbf{h}$.

The functions $f_\mathcal{I}$ and $f_\mathcal{S}$ are trained using batches of individual voxels. For a given voxel coordinate $\mathbf{c}_v$, functions $f_\mathcal{I}$ and $f_\mathcal{S}$ are tasked to predict intensity $\hat{i}_c$ and segmentation probabilities $\hat{\mathbf{s}}_c$ associated with that voxel. While CMR slices for a single subject are too sparse to accurately interpolate all inter-slice regions of a volume, this changes under cohort-wide training schemes. Variability across the cohort in slice positions and orientation relative to the standardized coordinate system allows the network $f_\psi$ to see all regions of the heart's anatomy across training. The network $f_\psi$ becomes capable of 'filling in' the imaging gaps of one subject from statistical population-wide patterns of those regions in world space.

Since the CMR image sequence precisely covers one full heartbeat, the time coordinate $t$ is mapped into cyclical representation via a sine-cosine value pair $(sin(\pi t), cos(\pi t))$. A sinusoidal positional encoder is then applied to the five coordinate values $[x, y, z, sin(\pi t), cos(\pi t)]$. Each spatial dimension is allocated seven frequencies, and each time dimension five, resulting in a vector of size 31.

**Geometry-grounded priors.** Learning directly from a coordinates in a standardized world space of anatomically aligned subjects has the benefit of building a strong positional bias of where to expect anatomical features. The network exploits statistical correlations on tissue positions and cardiac motion, forcing subject-specific features to be captured by a conditioning vector $\mathbf{h}_n$. The learnable vector $\mathbf{h}_n$ thus captures a compressed, implicit representation of how a specific subject $n$ deviates from the spatial prior learned by the network $f_\psi$. We implement this representation space $\mathcal{H}$ via a $N \times 128$ learnable matrix $\mathbf{H}$ initialized via $\mathcal{N}(0, 10^{-4})$, where each row $\mathbf{h}$ is 128-sized representation vector assigned to each of the $N$ training subjects. To prevent the training process from pushing the training representations $\mathbf{H}$ far apart and leaving unsupervised 'gaps' in the representation space $\mathcal{H}$, we enforce compactness in the space $\mathcal{H}$ by applying $L2$ regularization to both the network's weights $\psi$ and the representation matrix $\mathbf{H}$.

**Motion correction.** As CMR slices are acquired in separate breath-holds, it is commonplace for some amount of relative respiratory and subject motion to be introduced between images. To prevent the network from encoding these misalignments as subject-specific anatomical characteristics, we assume rigid motion and allocate learnable 3D rotation $\delta\theta$ and translation $\delta\tau$ offset vectors to each slice, regularizing their magnitude to mitigate parameter drifts. Each voxel belonging to the same slice is thus transformed into a standardized world-space via:

$$\mathbf{x}^{world} = \mathbf{R}(\theta + \delta\theta, \lambda)\mathbf{x}^{image} + \tau + \delta\tau$$

Here $\mathbf{R}$ is a rotation matrix composed from rotation angles $\theta$ and voxel spacings $\lambda$, and $\tau$ is a translation vector.

Because of the limited representational capacity of both the network $f_\psi$ and the representation space $\mathcal{H}$, motion-derived shape deviations from the implicitly-learned 'prototypical' heart structure can be resolved in an unsupervised manner by backpropagating gradients through the network. In an end-to-end fashion, this allows rotation and translation parameters of each slice's affine projection to be updated to more easily compress a given subject into its representation $\mathbf{h}$, and hence improve reconstruction fidelity. Unlike concurrent CMR motion-correction literature, our implicit approach does not suffer from the drawbacks of explicit statistical shape approaches, as well as does not require slices to intersect to compute intensity conflicts.

**Physics-informed intensity modeling.** In CMR imaging, the image intensity of a given point may vary across imaging planes depending on orientation. To allow our point-wise function $f_\psi$ to model orientation-dependent intensity variations, we apply the concept of a 'canonical view'. Under this notion, a given subject has a 'canonical' representation that is shared across all slices, but its intensities can be altered to match the observed intensities of any given voxel depending on the orientation of the source slice. We implement two key physically-inspired design choices to handle orientation-dependency: learnable slice-dependent intensity scaling and point-spread functions.

Inter-slice intensity variations exist due to receiver coil bias, respiratory drift between breath-holds, and flow-related CMR enhancement. To reconcile these incoherent 2D inputs into a consistent 3D representation, we optimize per-slice intensity scaling parameters

alongside the intensity function $f_{\mathcal{I}}$. We create a learnable matrix $D \in \mathbb{R}^{N \times J}$ for $N$ subjects and $J$ slices per subject. Each $d \in D$ scales all voxel intensities $\boldsymbol{I}_{[n,j]}$ of a particular slice $j$ of subject $n$ such that $(1+d) \cdot \boldsymbol{I}_{[n,j]} = I_{\text{canonical}}$, prior to supervising the predicted canonical view intensities $f_{\mathcal{I}}$. To prevent degenerate solutions, we regularize the magnitude $|d|$ to be near zero.

Next, we consider the issue of partial volume effects, whereby an observed voxel intensity is an accumulation of intensities of nearby tissues according to the scanner's resolution. The function modeling the spatial integration of signal intensities is termed the Point Spread Function (PSF) (Kuklisova-Murgasova et al., 2012; Rousseau et al., 2005). Following (Xu et al., 2023), we approximate the PSF as an anisotropic Gaussian function with a diagonal covariance matrix:

$$\boldsymbol{\Sigma} = \text{diag}\left(\left(\frac{r_x}{\kappa}\right)^2, \left(\frac{r_y}{\kappa}\right)^2, \left(\frac{r_z}{\kappa}\right)^2\right)$$

Here $\kappa = 2\sqrt{2 \ln 2} \approx 2.355$ relates the standard deviation to the Full-Width at Half-Maximum (FWHM), $r_x, r_y$ are the in-plane resolution, and $r_z$ the slice thickness.

Modeling voxel intensities thus corresponds to synthesizing low-resolution slices by performing an integral over the spatial components of $f_{\mathcal{I}}(\mathbf{x}, t, \mathbf{h})$. For a given voxel $v$, we approximate this integral via Monte-Carlo sampling using $K$ coordinates $\mathbf{x}^{(k)} \sim \mathcal{N}(\mathbf{x}_v, \boldsymbol{\Sigma}_j)$ such that the predicted intensity $\hat{i}_v$ is the following spatial average:

$$\hat{i}_v = \frac{1}{K} \sum_{k=1}^{K} f_{\mathcal{I}}(\mathbf{x}^{(k)}, t, \mathbf{h}) \approx \int f_{\mathcal{I}}(\mathbf{x}, t, \mathbf{h}) \mathcal{N}(\mathbf{x}; \mathbf{x}_v, \boldsymbol{\Sigma}_j) \, d\mathbf{x}$$

Due to the memory cost of back-propagating through all samples for a single voxel, the integral is approximated using $K = 16$ samples per voxel. We offer an ablation for PSF in Appendix C.

### 3.3. Optimization process

**Training.** Each training step samples the ground-truth intensities $i_v$ and segmentations $\mathbf{s}_v$ of 35000 voxels from a subject's 2D+time slice set. Intensities are scaled using their originating slice's learnable intensity scaling factor $d$. These serve as supervision to each voxel's predictions $\hat{i}_v$ and $\hat{\mathbf{s}}_v$. If voxels originate from a LA frame without annotations, the segmentation prediction is not supervised. The overall objective for a given voxel can be summarized as follows:

$$\begin{aligned}
\mathcal{L}_{total}(\psi, \mathbf{h}_n, \delta\theta_{nj}, \delta\tau_{nj}, d_{nj}) &= \mathcal{L}_{\text{PSNR}}(\hat{i}_v, i_v) + \mathcal{L}_{\text{Dice}}(\hat{\mathbf{s}}_v, s_v) + \alpha_{\text{weights}} \mathcal{L}_{L2}(\psi) \\
&+ \alpha_{\text{latent}} \mathcal{L}_{L2}(\mathbf{h}_n) + \alpha_{\text{rot}} \mathcal{L}_{L2}(\delta\theta_{nj}) + \alpha_{\text{tra}} \mathcal{L}_{L2}(\delta\tau_{nj}) + \alpha_{\text{intens}} \mathcal{L}_{L2}(d_{nj})
\end{aligned}$$

where $n$ and $j$ are the subject index and slice index corresponding to the randomly sampled voxel $v$. The network parameters $\psi$ are shared across all voxels, independent of which slice or subject a voxel originates from. We use $\alpha$ to denote weighting hyperparameters for the regularization components. We use $\alpha_{\text{weights}} = 10^{-5}$, $\alpha_{\text{latent}} = 10^{-4}$, $\alpha_{\text{rot}} = 10^{-4}$, $\alpha_{\text{tra}} = 10^{-4}$, and $\alpha_{\text{intens}} = 10^{-2}$. We train the architecture for 40000 epochs on an NVIDIA A100 GPU, lasting a total of 7 days.

**Inference.** To infer the 3D+t intensity volume and the corresponding segmentation for a new subject, a new latent vector, rotation, translation, and intensity scaling parameters are initialized as zeros, and are optimized exclusively on available image information (as we assume no test-time labels are available). These parameters are optimized by backpropagating gradients through $f_{\mathcal{I}}$ while keeping weights $\psi$ frozen for a total of 2500 steps (2 minutes). We accelerate this process by not using a PSF, as we empirically find no difference over using PSF sampling.

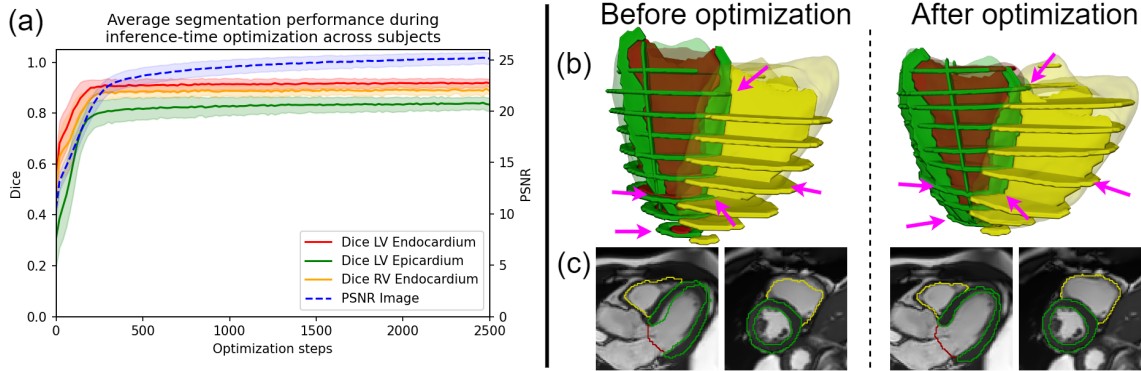

Figure 2: **(a)** Segmentation and reconstruction metrics during inference-time optimization on image intensities. Segmentation metrics plateau soon after the initial rough anatomy is modeled. **(b)** Motion between acquisitions causes the anatomy in slices to not correctly overlap in scanner space. Ground-truth slice segmentations before alignment clearly show errors in overlap. After optimizing rotation and translation parameters overlap is improved. **(c)** The original rotation and translation parameters show the cross-section of the predicted 3D segmentation does not align with the anatomy in the image, whereas after optimization the segmentation cross-section aligns with the correct tissues.

## 4. Results

**Inference performance.** Figure 2a shows the trend in segmentation and reconstruction metrics as inference-time optimization is performed on image intensities. While reconstruction performance keep climbing, after an initial period where subject anatomy is roughly represented, segmentation metrics reach an indefinite plateau. Interestingly, contrary to (Stolt-Ansó et al., 2023), segmentation metrics hardly deteriorate as optimization on intensities progresses. Table 1 shows the total segmentation and reconstruction scores, as well as separated by slice type.

Our approach appears to obtain values in-line with CMR segmentation literature (Bai et al., 2018). Apical regions are the exception, appearing to produce significantly worse metrics. We find our model produces plausible predictions, but our ground-truth segmentations frequently contain erroneous shapes. This is because our ground-truth is made up

of synthetic segmentations predicted by an in-plane 2D model (Bai et al., 2018) that lacks a holistic view of all imaging planes. A further discussion is offered in Appendix A, showcasing figures with examples of the unrealistic training shapes. We also provide reconstruction and segmentation results for hold-out tasks where representations are inferred using slice subsets (Appendix B).

Table 1: Test set reconstruction and segmentation scores across slice types. Dice scores are provided for foreground classes: LV blood pool (LV BP), LV myocardium (LV Myo), and RV blood pool (RV BP).

| Slice type | PSNR↑ | SSIM↑ | Dice (LV BP)↑ | Dice (LV Myo)↑ | Dice (RV BP)↑ |
|---|---|---|---|---|---|
| LA 2-ch | $23.80 \pm 1.86$ | $0.74 \pm 0.05$ | $0.93 \pm 0.02$ | $0.76 \pm 0.05$ | $0.79 \pm 0.37$ |
| LA 3-ch | $23.82 \pm 0.77$ | $0.79 \pm 0.02$ | $0.87 \pm 0.03$ | $0.78 \pm 0.05$ | $0.88 \pm 0.05$ |
| LA 4-ch | $24.62 \pm 0.71$ | $0.78 \pm 0.04$ | $0.93 \pm 0.02$ | $0.79 \pm 0.04$ | $0.91 \pm 0.02$ |
| SA basal | $25.12 \pm 0.77$ | $0.79 \pm 0.02$ | $0.87 \pm 0.18$ | $0.87 \pm 0.15$ | $0.85 \pm 0.19$ |
| SA mid-ventr. | $25.93 \pm 0.88$ | $0.81 \pm 0.03$ | $0.94 \pm 0.02$ | $0.87 \pm 0.04$ | $0.90 \pm 0.08$ |
| SA apical | $26.33 \pm 1.35$ | $0.83 \pm 0.04$ | $0.69 \pm 0.32$ | $0.58 \pm 0.31$ | $0.70 \pm 0.29$ |
| Average | $25.37 \pm 1.32$ | $0.80 \pm 0.04$ | $0.88 \pm 0.18$ | $0.81 \pm 0.18$ | $0.84 \pm 0.20$ |

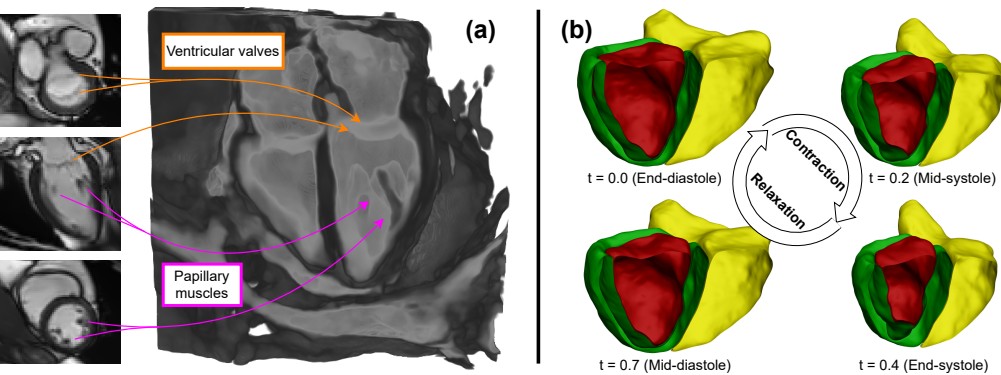

Figure 3: **(a)** Super-resolution intensity volume. For visualization purposes, highest intensities have been made transparent, allowing us to look into the ventricular and atrial cavities. Despite sub-structures such as papillary muscles and ventricular valves being only partially visible at any given image, our representation appears to understand their the continuous nature. **(b)** Meshes of super-resolution segmentation predictions. The representations display model smooth unbroken surfaces, even around the apical region, and accurately capture myocardial contraction and relaxation.

**Super-resolution capabilities.** Once a representation of a subject is optimized, any arbitrary coordinate can be queried, allowing us to sample volumes at much higher resolutions for any time point in the sequence. Figure 3 shows super-resolution results for a (300, 300,

300) resolution isotropic volume. The meshes generated from the super-resolution segmentation volume (Figure 3b) showcase our model's understanding of inter-slice regions beyond those of simple interpolation schemes, especially for regions suffering most from partial volume effects like apical and basal slices. Figure 3a shows a sliced intensity volume where high intensities have been made transparent, thus granting a view into ventricular and atrial cavities. Structures such as the papillary muscles and the ventricular valves, despite being only partially imaged on any given slice, are reconstructed by the model as continuous, unbroken structures.

**Motion correction.** As the inference process optimizes a representation, the rotation and translation parameters are updated alongside the latent code so as to best minimize the reconstruction error. This reduces intensity dissimilarities along slice intersections, as well as fits the anatomical features in those slices to the architecture's implicit prior learned over the training cohort. Figure 2b qualitatively highlights the ground-truth segmentation alignments before and after the optimization process.

Table 2: Differences in ground-truth intensities and segmentations along slice intersection before and after optimization of rotation and translation parameters in test set.

| Slice intersection intensity difference (NCC ↑) | | Slice intersection segmentation difference (Dice ↑) | |
|---|---|---|---|
| Before optimization | After optimization | Before optimization | After optimization |
| $0.521 \pm 0.108$ | $0.706 \pm 0.093$ | $0.756 \pm 0.016$ | $0.862 \pm 0.034$ |

Table 2 quantifies the improvement along slice intersections before and after alignment, both in terms of intensity differences (using normalized cross-correlation) and segmentation class differences (using Dice). It should be noted that because slices are annotated independently of each other, ground-truth segmentations never perfectly align in 3D even if we were to assume perfect alignment. Despite this view-dependent variability, our architecture produces a shared 3D representation that best fits all slices (see Figure 3). Figure 2c visualizes how the cross-section of the predicted 3D segmentation does not properly align with the anatomy on the original imaging planes. After optimization of rotation and translation parameters, the 3D segmentation cross-section fits the anatomy correctly along all views.

Figure 4 further emphasizes the motion correction capabilities by displaying intensities sampled along slice intersections before and after optimization. Before optimization, intensities along slice intersections often do not properly to match across views. However, after optimization, anatomical landmarks appear to be spatially consistent along slice intersections of both views.

## 5. Discussion

**Limitations.** A general trade-off in implicit representation approaches is between reconstruction fidelity and interpolation accuracy. The reliance of this work on image compression, while displaying remarkable interpolative properties for modeling general anatomical structures, leads to the loss of the fine-grained detail of the original slices. While dataset

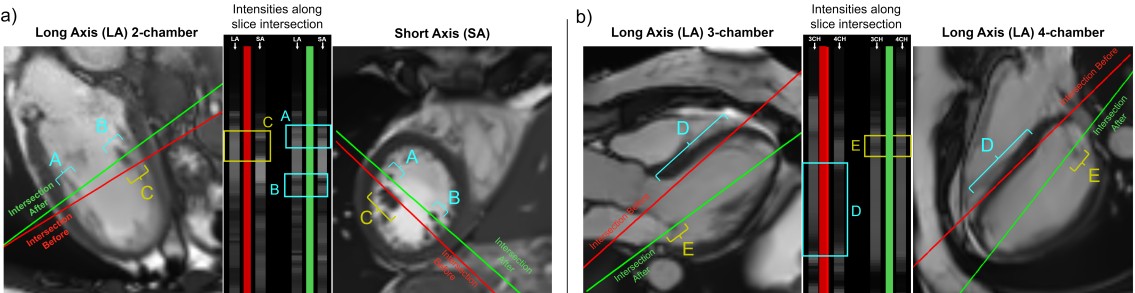

Figure 4: Motion correction results displaying intensities along the intersection of two slices. Red line indicates intersection of slices prior to optimization, while the green line is the post-optimization intersection. Sampled intensities along both lines are displayed in the vertical section between each pair, showing how anatomical landmarks become spatially consistent across slice pairs. Letters A-E indicate landmarks that appear to match in the post-optimization alignment.

and model size could be scaled up, there is a limit to the information that can be inferred from the standard set of thick-slice acquisitions of clinical CMR protocols.

The reference segmentations used were automatic outputs of a trained 2D model (Bai et al., 2018) that lacks a holistic view of all imaging planes. Even though implicit representations functionally 'smooth-out' outlier predictions (Banus et al., 2025), it is unclear what general trends in these faulty segmentations our representations extract. This is most important to consider in apical regions, where the frequency of erroneous segmentations by these convolutional architectures are substantial (see Appendix A).

Additionally, a general trend in the implicit representation literature is the requirement of long training times. While our training time of 7 days is in line with adjacent works (9 days (Stolt-Ansó et al., 2023), 4 days (Banus et al., 2025)), this limits the potential for widespread adoption. However, we believe there are promising research avenues to reduce this cost, such as via smarter sampling strategies (Tack et al., 2023) or by employing simpler primitives beyond neural networks (Zhang et al., 2025).

**Future work.** The ability to generate high-resolution intensity volumes opens up possibilities of using 3D annotation tools to achieve view-consistent segmentations. We see this as especially relevant for studying function of regions of the heart whose shape modeling is currently off-limits due to the inability to produce detailed view-coherent segmentations from the sparse partial views.

Additionally, recent works have highlighted the benefits of shape modeling via signed distance functions, as opposed to occupancy functions (Gropp et al., 2020; Kuipers et al., 2025). Signed distance functions approaches offer stronger inductive biases (eg. Eikonal constraints) and do not require engineering efforts associated with handling partially annotated data.

Other promising future research avenues include the use of shape templates from high-resolution data (Sander et al., 2023), using more sophisticated conditioning mechanisms (Perez et al., 2018), and improving spatial resolution via multi-resolution latent grids (Bauer et al., 2023; Müller et al., 2022).

## 6. Conclusion

This work has explored the use of implicit representations to combine all available imaging slices of standard CMR acquisitions and create shared representations capable of producing super-resolution intensity and segmentation volumes. Our approach involves physics-inspired techniques to account for view-dependent intensity differences across slices. The optimization process of these representations additionally corrects for rigid motion between acquisition planes. We show that despite suboptimal ground-truth segmentations, implicit representations produce coherent global segmentations by integrating information across all views.

## Acknowledgments

This work is funded by the Munich Center for Machine Learning. This research has been conducted using the UK Biobank Resource under Application Number 87802.

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

## Appendix A. Issues with apical training labels

While we employ synthetic labels as segmentation ground-truth due to the benefit of having predictions at every time point, this introduces the potential issue of training on inaccurate labels. In this work, we use a trained CNN provided by (Bai et al., 2018), which is slice-specific and thus employs no context from other slices. We notice this CNN to commonly fails to predict realistic segmentations for apical SA slices. We suspect the reason for this is two-fold. First, the CNN lacks context to precisely track through-plane motion. Second, the training of this network suffered from shape imbalance, as the majority of SA slices in a stack are mid-ventricular, which consist of a LV blood pool with a myocardium 'ring' around the blood pool. We notice the majority of CNN-predicted apical segmentations contain erroneous LV shapes. Figures 5, 6, 7 showcase common errors found in our apical training labels, mostly consisting of incorrect LV shapes.

Interestingly, as seen in Figures 5, 6, 7, our method manages correctly segment these regions despite the presence of these types of error in its training set. We believe this helps explain the large drop in segmentation Dice scores observed in the results in Table 1. We hypothesize two reasons on why our method may be overcoming these errors in its training data. First, the inclusion of long axis views contributes to the supervision of these regions, and since there are three LA views but only one apical SA slice, the LA views provide a stronger signal towards determining what kind of shape the apical region predominantly produces. Our experiments on inference using only LA views in Appendix B further reinforce this hypothesis. The second reason may be due to the compressive nature of implicit representations. As pointed out by (Banus et al., 2025), outliers in the data are 'smoothed' away during the representation-building process. If apical errors occur erratically, our architecture might not be capable to fully minimize the segmentation loss by learning to reliably reconstruct these incorrect shapes.

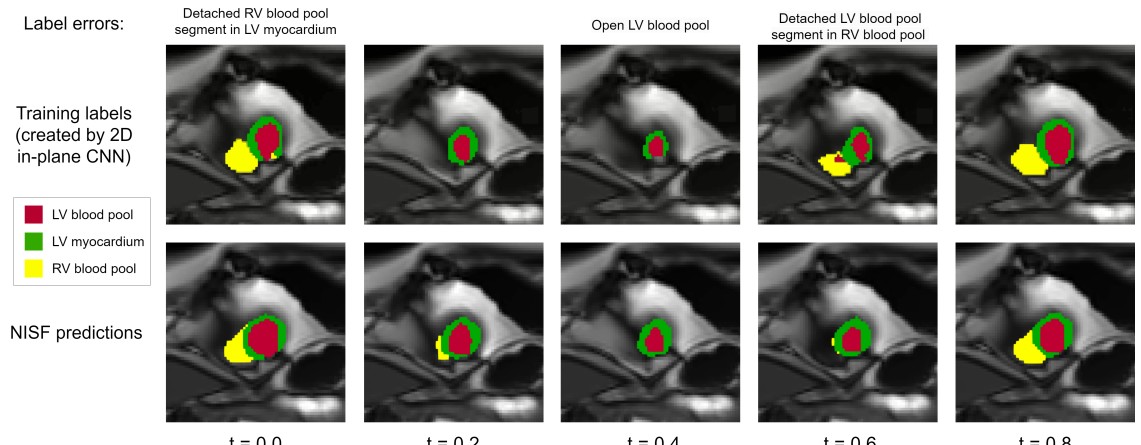

Figure 5: Top rows displays our synthetic ground-truth training data. These are predicted by a 2D CNN, resulting in frequent segmentation errors in apical regions. The LV myocardium appears to have holes, and detached segments appear inside other class labels. The bottom row shows cross-sectional slices of our model's 3D predicted segmentation, which portrays a more realistic shape of the ventricles despite the erroneous training data.

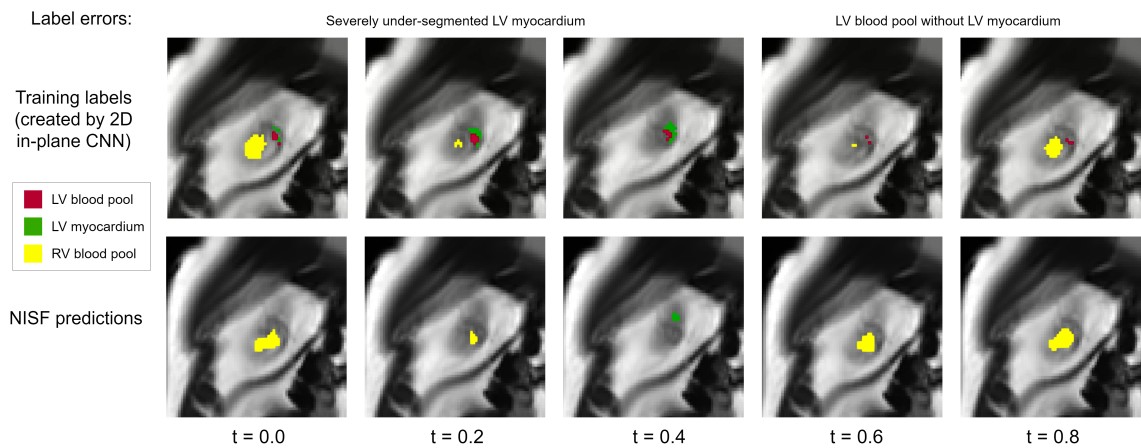

Figure 6: Top rows displays our synthetic ground-truth training data. These are predicted by a 2D CNN, resulting in frequent segmentation errors in apical regions. The LV myocardium appears to be severely under-segmented. The bottom row shows cross-sectional slices of our model's 3D predicted segmentation, which portrays a more realistic shape of the ventricles despite the erroneous training data.

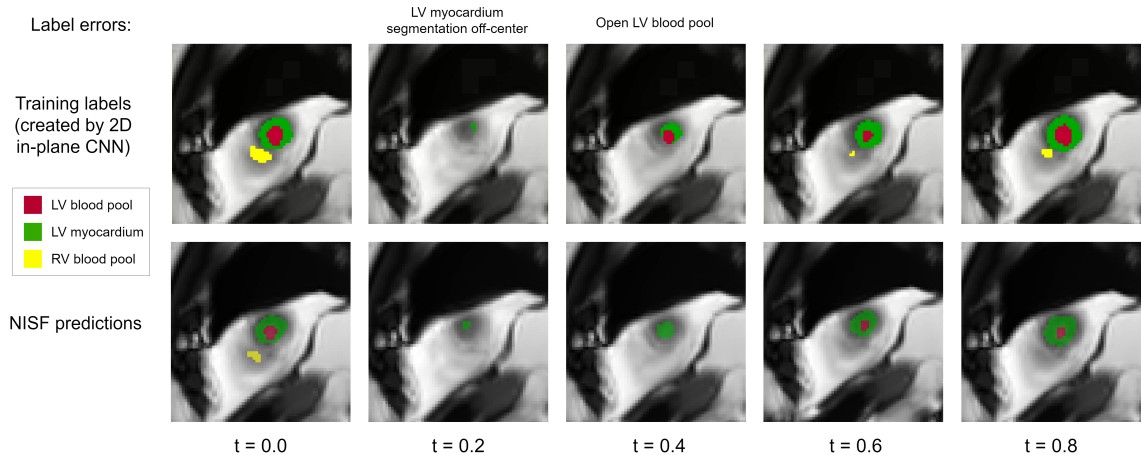

Figure 7: Top rows displays our synthetic ground-truth training data. These are predicted by a 2D CNN, resulting in frequent segmentation errors in apical regions. As the apex moves through the imaging plane, the LV myocardium segmentation is spatially inconsistent across time. As the apex crosses the imaging plane again, the LV myocardium is undersegmented, failing to enclose the LV blood-pool. The bottom row shows cross-sectional slices of our model's 3D predicted segmentation, which portrays a more realistic motion of the ventricles despite the erroneous training data.

## Appendix B. Reconstruction of missing acquisition planes

In this section, we investigate our architecture's ability to create coherent representations from incomplete sets of slices and reconstruct missing views. We use a model trained on all available SA and LA views and perform inference on new subjects using a subset of the available views.

### B.1. Building representation exclusively from SA slices

We restrict the inference optimization to only use voxel intensities originating from SA slices. After obtaining a representation from the SA images, we sample along the held-out LA views. Figure 8 displays the reconstructed intensities of the held-out LA images, as well as the generated segmentations for those held-out views.

The reconstructed images, while lacking fine-grained detail, are able to capture the subject-specific anatomy of the held out images strikingly well. While there are some differences in the shapes of the generated segmentations, it is unclear how much of this can be also attributed to drift in the motion-corrected parameters. Also, the apical regions showcase undersegmented myocardium, leaving the LV blood pool exposed. We suspect this may be caused by the SA views having little-to-no information on how the apical region of the LV myocardium behaves, causing the intensities of that region to not be properly reconstructed and exposing the LV blood pool through the apex.

### B.2. Building representation exclusively from LA slices

Next, we instead restrict the inference optimization to only use voxel intensities originating from three LA slices. After optimizing a representation from these three LA images, we sample along the planes of the held-out SA stack. Figure 9 displays the reconstructed intensities of a held-out SA basal slice, a mid-ventricular slice, and a near-apical slice, as well as the generated segmentations for these held-out views.

Similarly to the previous section, the reconstructed images lack fine-grained detail, but are able to capture the subject-specific anatomy of the held out images strikingly well. While the ground-truth segmentations tend to be quite circular, the LA-derived SA segmentations are slightly less round, appearing to deform outwards at the intersection of the LA planes.

A note-worthy detail about LA-derived representations is that apical SA views appear to correctly model the through-plane motion of the LV myocardium. Figure 10 highlights such an example of an apical view. The 3D segmentation is generated by a LA-derived representation, we take the cross-sectional slice for that specific apical view. As we sample this slice along the time dimension, we observe anatomically correct through-plane motion. First, we see the LV blood pool moving out-of-plane, leaving only the myocardium visible, and later reappearing. This is not the case for the synthetic ground-truth segmentations (which are generated by a 2D CNN). This reinforces our hypothesis that the source of errors in apical predictions are caused by the commonly erroneous training data of SA apical segmentations.

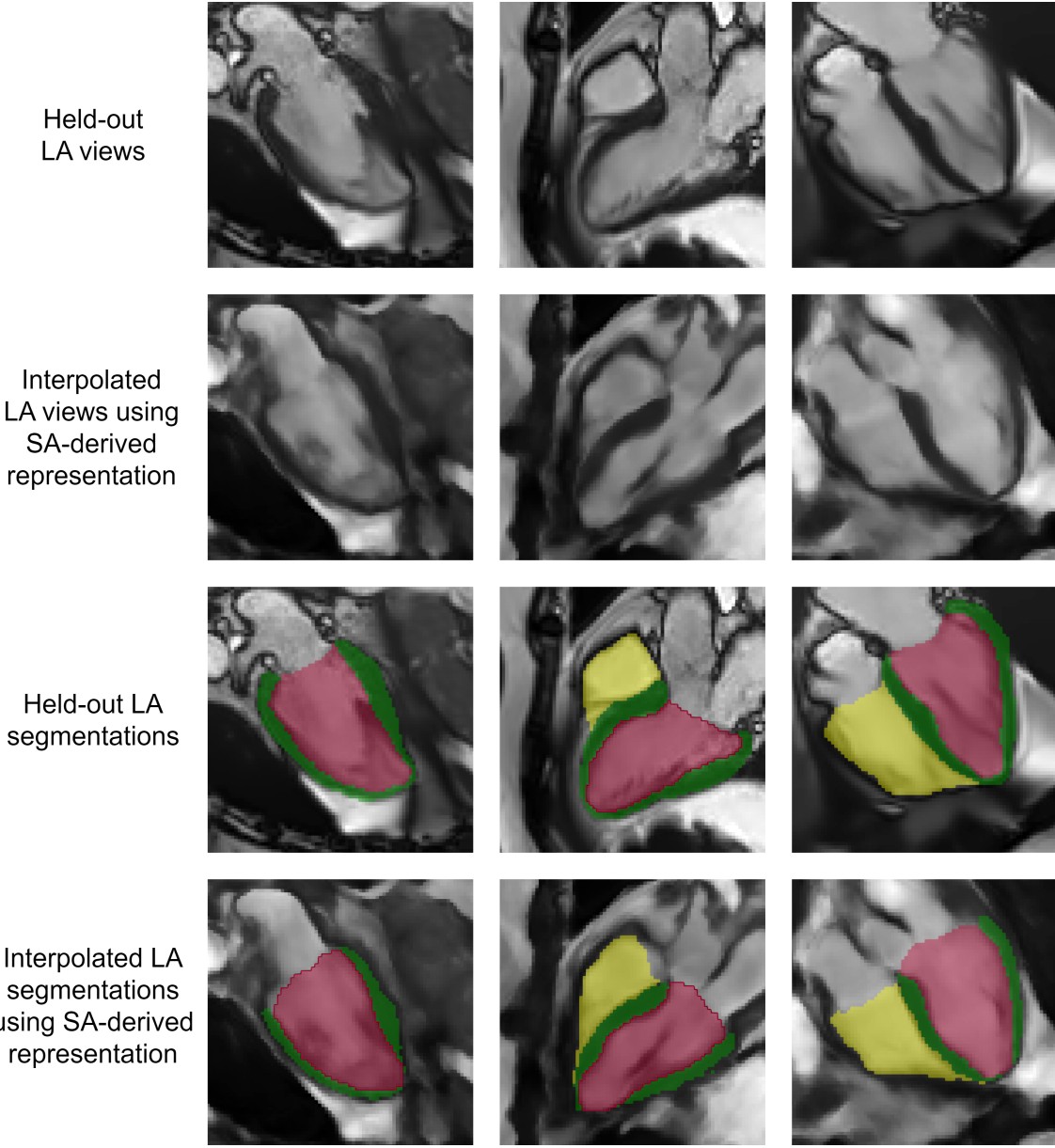

Figure 8: The top row shows the original held-out ground-truth LA images. The second row displays the reconstructed LA images from the representation derived from optimizing on SA images. The third row shows the ground-truth segmentations overlayed on the orignal LA images. The last row shows the generated LA segmentations from the representation derived from the SA images.

Held-out
SA views

Interpolated
SA views using
LA-derived
representation

Held-out SA
segmentations

Interpolated SA
segmentations
using LA-derived
representation

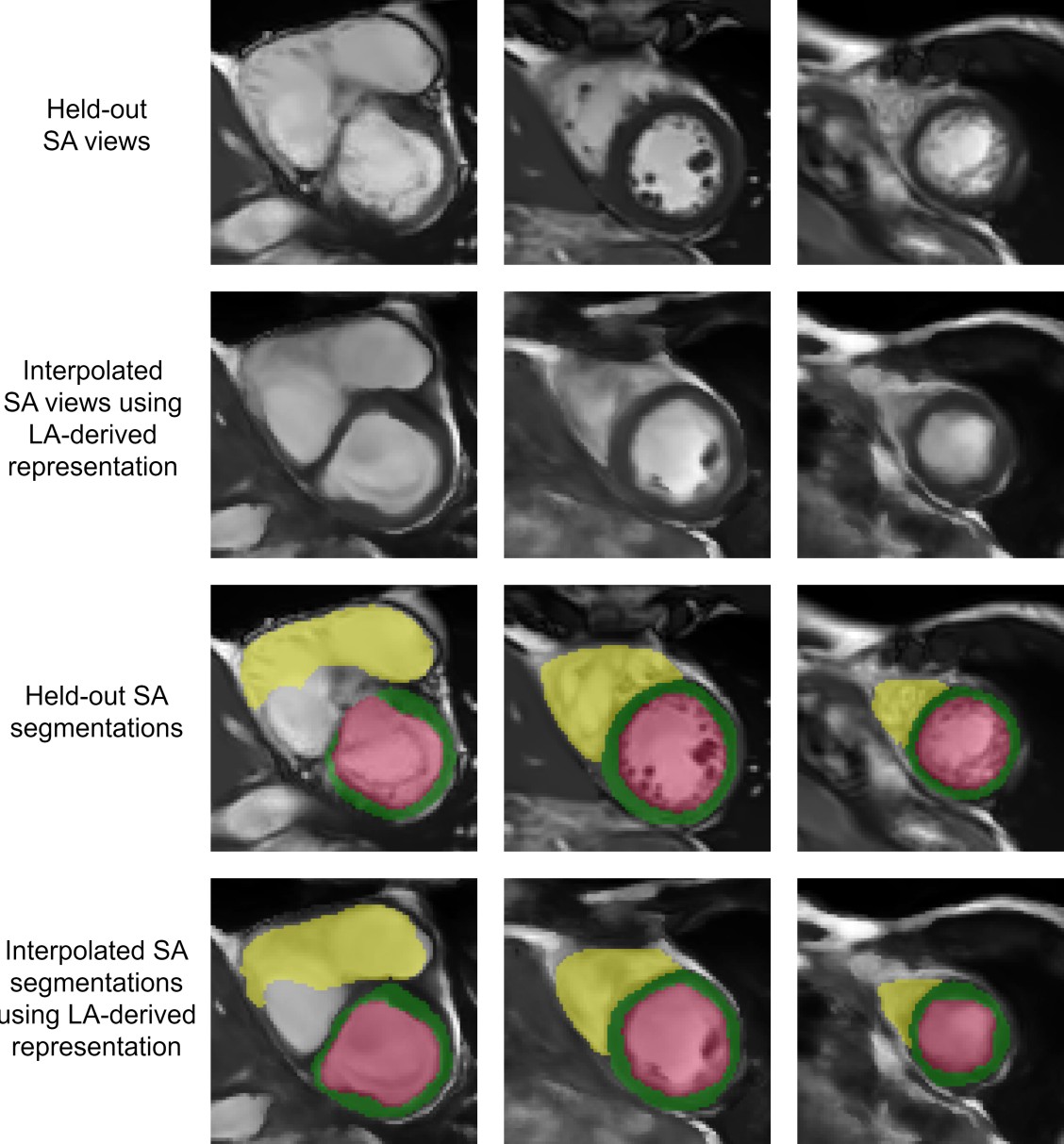

Figure 9: The top row shows the original held-out ground-truth SA images. The second row displays the reconstructed SA images from the representation derived from optimizing on LA images. The third row shows the ground-truth segmentations overlayed on the original SA images. The last row shows the generated SA segmentations from the representation derived from the LA images.

Held-out
apical SA
segmentations

Interpolated apical SA
segmentations using
LA-derived
representation

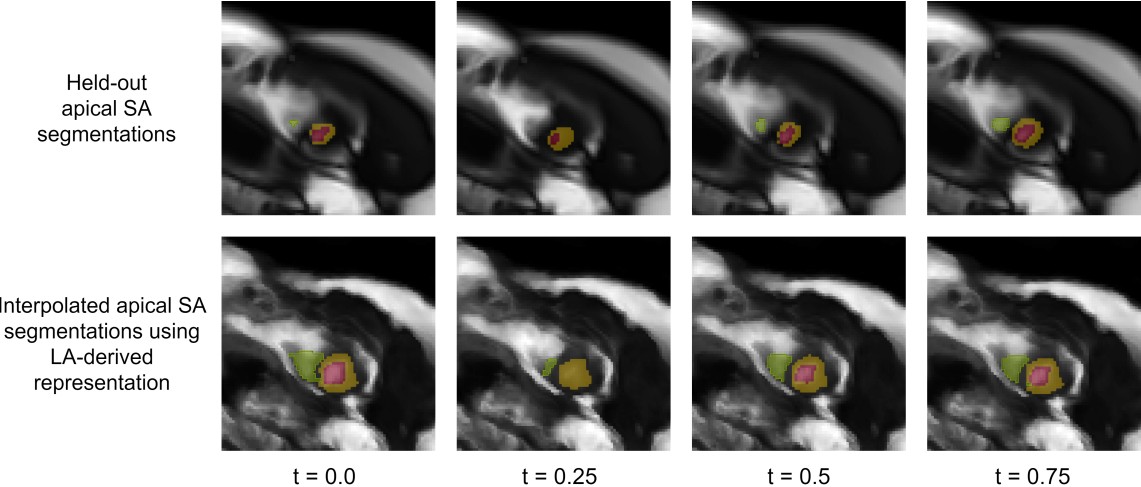

t = 0.0          t = 0.25          t = 0.5          t = 0.75

Figure 10: Segmentation of an apical slice over multiple time points. Top row is the ground-truth segmentation. Bottom row is the segmentation generated using a LA-derived representation, which follows accuracte through-plane motion.

## Appendix C. PSF ablation

The use of PSF, while being commonplace in implicit representation literature for magnetic resonance imaging, is not obvious for the CMR domain due to the low number of overlapping sections between slices. Despite adding a significant cost in memory requirements, we find using a PSF provides a crucial form of regularization. Figure 11 highlights the intensity artifacts produced in the setting lacking PSF. Banding artifacts are present along slice intersections. In LA images, the SA intersections are clearly visible, as the network appears to predict different intensities along where the SA slices intersect with the LA plane. Similarly, the reconstruction of the mid-ventricular SA slice appears to have bright line artifacts along where the LA images intersect. Adding sampling using the PSF improves image clarity and removes the artifacts.

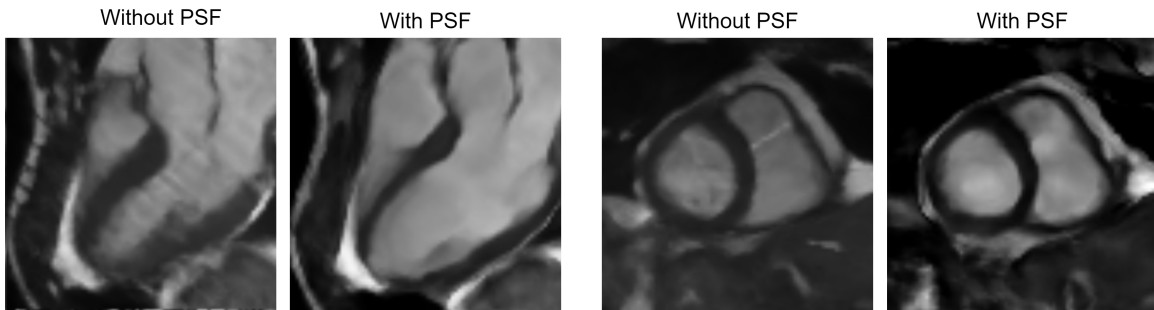

Figure 11: In-plane reconstruction of a LA 3-chamber view (left) and a mid-ventricular SA view (right). When no PSF is used, banding artifacts occur along slice intersections. Using PSF removes these artifacts and improves reconstruction clarity.

