# OpenReview forum: "NISF++: Geometrically-grounded implicit representations of 3D+time cardiac function from 2D short- and long-axis MR views"
_MIDL.io/2026/Conference — MIDL 2026 Poster_

### Official Review · Reviewer_1PVa · 2025-12-29

**Confidence:** 4
**Preliminary Rating:** 4
**Final Rating:** 5

**Summary:**

This paper proposes NISF++, an extension of neural implicit segmentation functions for cardiac MR imaging that builds 3D+time representations from multi-view 2D CMR acquisitions (short-axis and long-axis views). The method uses an auto-decoder architecture with learnable subject-specific latent codes $\mathbf{h} \in \mathbb{R}^{128}$, mapping spatiotemporal coordinates $\mathbf{c} = (x, y, z, t) \in \Omega \subset \mathbb{R}^4$ to intensity and segmentation fields via a shared MLP $f_\psi(\mathbf{c}, \mathbf{h}) = (f_I, f_S)$. Key contributions include rigid motion correction via learnable per-slice rotation $\delta\theta$ and translation $\delta\tau$ parameters, physics-informed intensity modeling using point spread functions, and super-resolution capabilities. The method is evaluated on 120 UK Biobank subjects (100/10/10 split), demonstrating segmentation performance the authors claim is on-par with existing methods, improved slice alignment after motion correction, and qualitatively compelling super-resolution results. The work addresses the clinically relevant challenge of integrating sparse, heterogeneously-oriented CMR slices into coherent spatiotemporal representations.

**Strengths:**

**S1. Clinically relevant problem formulation.** The challenge of integrating sparse, multi-orientation 2D CMR slices into coherent 3D+time representations is genuinely important. Standard deep learning architectures cannot naturally handle heterogeneous slice orientations in clinical CMR protocols.

**S2. Elegant unified framework.** The approach handles multiple tasks (segmentation, reconstruction, motion correction, super-resolution) within a single optimization framework. Treating all CMR slices as a point cloud in standardized world coordinates is intuitive and well-motivated.

**S3. Principled handling of CMR-specific challenges.** The physics-informed components (PSF modeling, per-slice intensity scaling) address real acquisition-related variations. The PSF ablation (Appendix C) demonstrates these components prevent artifacts.

**S4. Interesting generalization experiments.** Appendix B shows representations can be built from SA-only or LA-only views and generalize to held-out orientations, demonstrating meaningful learned anatomical priors.

**S5. Transparent discussion of limitations.** The authors honestly discuss ground truth quality issues in apical regions and the reconstruction-interpolation trade-off.

**Weaknesses:**

**W1. Geometrically-groundedness in title is overstated.** The term appears only in the title and one subsection header. The method contains no geometric grounding in the accepted sense—no signed distance functions, Eikonal constraints, surface normal supervision, curvature regularization, equivariant constraints, or topological guarantees. The described "geometry-grounded priors" are population-level spatial statistics in canonical coordinates, which is coordinate-based learning, not geometric grounding, and has been proposed before [1].

**W2. Occupancy vs SDFs.** The authors cite DeepSDF as foundational but abandon its core representation (signed distance functions) for occupancy fields without justification. SDFs would provide inherent geometric properties and could naturally handle nested cardiac anatomy (LV blood pool $\subset$ myocardium) through hierarchical distance fields.

**W3. No topological consistency evaluation.** Despite identifying topological errors (holes, disconnected components) as key failure modes of existing methods, the authors never quantify whether their approach produces topologically valid outputs. Cardiac segmentations require strict constraints (single connected components, watertight surfaces, proper nesting) that are neither measured nor guaranteed.

**W4. Small test set.** Only $n=10$ test subjects is statistically inadequate. Standard deviations are often 20-40% of means (e.g., apical LV myocardium Dice: $0.58 \pm 0.31$). No confidence intervals or significance tests are reported.

**W5. Circular ground truth problem.** Training segmentations from Bai et al. (2018) serve as ground truth against which performance is compared. The claim of "better anatomical accuracy than training data" relies solely on visual inspection without expert validation.

**W6. Missing baseline comparisons.** No comparisons to NIMOSEF (Banus et al., 2025)—a direct competitor cited in the paper—nor to Zhang et al. (2024), explicit motion correction methods, or super-resolution baselines.

**W7. Purely qualitative super-resolution evaluation.** No quantitative metrics are reported; claims about papillary muscle and valve reconstruction are unvalidated.

[1] Sander, J., de Vos, B. D., Bruns, S., Planken, N., Viergever, M. A., Leiner, T., & Išgum, I. (2023). Reconstruction and completion of high-resolution 3D cardiac shapes using anisotropic CMRI segmentations and continuous implicit neural representations. Computers in Biology and Medicine, 164, 107266.

**Detailed Comments:**

### On the "Geometrically-Grounded" Claim

In the implicit representation literature, geometric grounding typically involves SDFs with Eikonal constraints ($\|\nabla f\| = 1$), surface normal supervision, or topology-aware losses. NISF++ contains none of these. The "Geometry-grounded priors" subsection describes learning spatial statistics in standardized coordinates, which is valuable, but not geometric grounding. A more accurate title would reference "coordinate-based" or "population-prior-informed" representations.

### On the Representation Choice

The occupancy vs. SDF choice has significant implications:

| Aspect | Occupancy | SDF |
|--------|-----------|-----|
| Surface normals | Not available | $\nabla f$ |
| Topological guarantees | None | Possible with Eikonal |
| Multi-class nesting | Independent | Natural hierarchy |

For cardiac anatomy with strict nesting constraints, hierarchical SDFs would be natural.

### On Evaluation

The evaluation has interconnected problems: (1) with $n=10$ and high variance, conclusions are unreliable; (2) using Bai et al. predictions as ground truth measures model agreement, not anatomical accuracy; (3) without baselines, added complexity (7 days training) cannot be justified.

### Minor Issues

- Typos: "wholistic" → "holistic"; "cohorent" → "coherent"
- Missing details: PSNR loss baseline, Dice loss variant, PSF sample count

**Justification Of Final Rating:**

Though some final disagreements remain which I think merit discussion, I think this work is a valuable contribution to MIDL. Besides this, the authors have actively responded in the Discussion period, which I much appreciate. I am therefore happy to promote this work with a strong accept as a final decision.

**Justification Of The Preliminary Rating:**

The paper tackles a clinically relevant and challenging problem—integrating sparse, multi-orientation 2D CMR slices into coherent 3D+time representations—with an elegant unified framework well-suited for the MIDL community. The core contribution of using neural implicit functions to jointly handle segmentation, motion correction, and super-resolution from heterogeneous CMR views is novel and technically sound. The physics-informed components (PSF modeling, per-slice intensity scaling) demonstrate careful consideration of domain-specific challenges, and the generalization experiments (Appendix B) provide compelling evidence of meaningful learned anatomical priors.

While there are valid concerns, these do not fundamentally undermine the paper's contributions. The method elegantly addresses a real clinical need, the qualitative results are convincing, and the framework opens interesting directions for future work (e.g., modeling pathological hearts, non-rigid motion). The transparent discussion of ground truth limitations and failure cases reflects scientific rigor.

For MIDL, where novel ideas and potential are valued alongside rigorous benchmarking, this paper makes a solid contribution. The authors should clarify the "geometrically-grounded" framing and consider adding topology metrics, but the core work merits presentation and discussion at the conference.

**Questions To Address In The Rebuttal:**

**Q1.** The title claims "geometrically-grounded" representations, but the method contains no geometric constraints. Can you clarify what "geometrically-grounded" means in your context and how it differs from standard coordinate-based INRs?

**Q2.** Why did you choose occupancy fields over signed distance functions, despite citing DeepSDF? SDFs could naturally encode nested cardiac anatomy and provide surface normals (also see [1]). Was this empirically motivated?

**Q3.** You identify topological errors as failures of existing methods and claim improved anatomical accuracy. Can you provide quantitative topology metrics (connected components, watertightness, containment violations)?

**Q4.** The claim that your method outperforms training labels in apical regions is based on visual inspection. Can you provide expert validation or quantitative evidence?

**Q5.** Why are no baseline comparisons included, particularly to NIMOSEF (Banus et al., 2025)?

**Q6.** Can you provide quantitative super-resolution evaluation?

---

> ### Author Response · Authors · 2026-01-24
> **Geometrically-groundedness, baselines, super-resolution metrics  (part 1)**
>
> We thank the reviewer for their positive assessment of our paper, calling it a “clinically relevant problem formulation”, and appreciate the detailed evaluation of the paper. We subsequently offer clarification on the reviewer’s questions and comments:
>
>
> ## Addressing claims on geometric-groundedness
>
> The reviewer makes a valid claim in that standard neural implicit approaches simply perform coordinate-based learning and carry no geometrical constraints. This is indeed the case for approaches such as NISF [4] or NIMOSEF [5] whereby subject-level differences in position and orientation are learned as features in the latent space.
> However, we disagree that this is also the case in our work. In allowing our slices to learn rigid transformation parameters, we introduce a SE3 geometry to our learning scheme. This opens the learning process to encode acquisition artifacts such as inter-slice motion or global translations/rotations, no longer as subject-specific features, but as slice-specific transformations. This is a similar claim to those made in popular concurrent works [6].
>
>
>
> ### Occupancy vs signed distance field modeling:
>
> We thank the reviewer for this interesting suggestion. While we believe a SDF-based representation to be a conceptually promising direction, allowing to integrate Eikonal -/ geometrical constraints (similar to Amos et al.) [1], it is unfortunately not possible to integrate this into the existing setup based on two distinct points:
>
> (1) absence of ground truth data. While signed distances can be calculated in-plane, the sparsity of slices in the CMR domain makes it impossible to know the true distance to the surface of the underlying surface. This problem is exacerbated by the presence of motion across slices, making it impossible to derive the true distance to the contours of nearby slices.
>
> (2) SDFs are much harder to optimize and typically require hours to optimize for single shapes [2,3]. This is particularly evident in [2,3] as sampling strategies are an active research area.
>
> We would also like to comment on the reviewer’s criticism of our citation of DeepSDF. We would argue DeepSDF is predominantly known in object modelling research for being the first to propose a cohort-wide training scheme with an auto-regressive architecture. Other known works cite DeepSDF for its architecture despite not using signed distance fields [7].
>
> On the claims that the 7-day training scheme is unjustifiable, we respectfully disagree with the reviewer. We refer the reviewer to the general rebuttal section where we report training times listed in adjacent research papers and discuss contributing factors.
>
>
>
> ## Baseline comparisons and NIMOSEF (Banus et al., 2025)
>
> We would like to refer you to our general response which provides an intuition regarding our baseline setup. In short, to the best of our knowledge, no other work addresses the shape modeling task our work explores. In-plane segmentation baselines solve a different task than our approach does. Our hope is to have readers understand the limitations associated with CMR imaging and annotation data. We believe baselines that optimize for in-plane segmentation might mislead readers into believing the two tasks are equivalent.
>
> We would also like to point out that NIMOSEF does **not** offer a line of work comparable to ours. NIMOSEF is a simple extension to the original NISF [4], whereby an additional output head predicts tissue deformation along the temporal dimension. Just like NISF, NIMOSEF does not offer any form of slice motion correction and can only handle a static anisotropic short-axis stack. The validity of its contributions has also been questioned in the same line of reasoning as to why signed distance ground-truth aren’t obtainable in CMR data: the deformation ground-truths they supervise with are derived from in-plane motion. This ignores the fact that actual cardiac tissue motion is through-plane, casting uncertainty to the validity of what their deformation functions actually model. The work also exclusively uses a training set and does not perform inference on new subjects, neither does it make its deformation training data available nor offers details on how to reproduce it.
>
> As for Zhang et al. (2024), we exhaustively searched for previous works under that (unfortunately common) author name and publishing date, and were unable to determine what previous work the reviewer was referring to given the lack of a title.

---

> > ### Author Response · Authors · 2026-01-24
> > **(part 2)**
> >
> > ## Providing quantitative super-resolution evaluation
> >
> > Since it is not possible to obtain high resolution data (further complicated by motion artifacts inherent to CMR data), it is unfortunately not possible to provide quantitative scores or metrics for the super-resolution task. The reviewer points out the option to report topological metrics such as single connected components, watertight surfaces, and proper nesting. Because of the nature of ventricle occupancy labels, watertightness and proper nesting are not properties that apply to our task. The left ventricle’s myocardial tissue does not fully encompass its blood-pool due to the connectedness of the ventricle to the left atrium and the aorta. We do however see the benefit of reporting connected-component metrics. Furthermore, as discussed in the paper, the flaws of the (pseudo-) ground-truth training labels in the apical region causes our representations to not always exhibit closed apical regions, which is an important property for clinical-oriented research.
> > We are planning to expand on the utility in a follow-up study by evaluating its impact with clinical readers.
> >
> >
> > Please let us know whether our rebuttal effectively addresses all your questions and comments. We are ready to provide any additional clarification that may be needed.
> >
> > ### Citations:
> >
> > [1] Gropp A, Yariv L, Haim N, Atzmon M, Lipman Y. Implicit geometric regularization for learning shapes. arXiv preprint arXiv:2002.10099. 2020 Feb 24.Ategies
> >
> > [2] Davies T, Nowrouzezahrai D, Jacobson A. On the effectiveness of weight-encoded neural implicit 3d shapes. arXiv preprint arXiv:2009.09808. 2020 Sep 17.
> >
> > [3] Lindell DB, Van Veen D, Park JJ, Wetzstein G. Bacon: Band-limited coordinate networks for multiscale scene representation. InProceedings of the IEEE/CVF conference on computer vision and pattern recognition 2022 (pp. 16252-16262).
> >
> > [4] Stolt-Ansó N, McGinnis J, Pan J, Hammernik K, Rueckert D. Nisf: Neural implicit segmentation functions. InInternational Conference on Medical Image Computing and Computer-Assisted Intervention 2023 Oct 1 (pp. 734-744). Cham: Springer Nature Switzerland.
> >
> > [5] Banus J, Delaloye A, M. Gordaliza P, Georgantas C, van Heeswijk RB, Richiardi J. NIMOSEF: Neural Implicit Motion and Segmentation Functions. InInternational Conference on Medical Image Computing and Computer-Assisted Intervention 2025 Sep 20 (pp. 444-454). Cham: Springer Nature Switzerland.
> >
> > [6] Park K, Sinha U, Barron JT, Bouaziz S, Goldman DB, Seitz SM, Martin-Brualla R. Nerfies: Deformable neural radiance fields. InProceedings of the IEEE/CVF international conference on computer vision 2021 (pp. 5865-5874).
> >
> > [7] Amiranashvili T, Lüdke D, Li HB, Menze B, Zachow S. Learning shape reconstruction from sparse measurements with neural implicit functions. InInternational Conference on Medical Imaging with Deep Learning 2022 Dec 4 (pp. 22-34). PMLR.

---

> ### Comment · Reviewer_1PVa · 2026-01-26
> **Response to Rebuttal**
>
> I thank the authors for their thoughtful rebuttal. I'd like to continue the discussion on one point that I believe warrants further consideration, acknowledging that we are in the discussion phase and I am not expecting new experiments.
>
> **On self-supervised SDF learning**: The authors state that SDFs are impractical because ground truth signed distances cannot be computed from sparse CMR slices. However, I'd like to draw attention to recent work on self-supervised SDF learning that may be relevant here.
>
> Kuipers et al. (MIDL 2025) demonstrate that SDFs can be learned without ground truth distance values by leveraging the Eikonal constraint as an inductive bias. Their approach requires only:
> - Surface points (where d=0)
> - Eikonal regularization (||∇f|| = 1)
> - Optionally, surface normals
>
> This builds on Gropp et al. (2020) and has been applied to vascular structures by Alblas et al. (STACOM 2022). The key insight is that the SDF's mathematical properties allow self-supervision from boundary information alone.
>
> Given that NISF++ already has access to segmentation boundaries (surface points), this approach would seem applicable. I'd be curious to hear the authors' thoughts on whether self-supervised SDF learning was considered, and if there are domain-specific reasons it might not transfer to the cardiac MR setting.
>
> **On Topology** I accept the point regarding the LV blood pool's anatomical connections. However, the LV myocardium itself is a coherent muscular structure where connected component analysis remains meaningful. This would complement the existing evaluation nicely, though I understand it cannot be added at this stage.
>
> Overall, I appreciate the authors' engagement with my concerns, which are mostly addressed. The core contribution—multi-view CMR integration via implicit representations—is valuable, and this discussion is intended constructively to explore the methodological design space.

---

> > ### Author Response · Authors · 2026-01-29
> > **Response to SDF learning and topology**
> >
> > We would like to thank the reviewer for the thorough comments. We very much appreciate the clarification on the SDF learning and agree with the reviewer’s points. We were not aware of Eikonal regularization and are genuinely curious to try it on our future projects. Previous attempts of ours at naively using SDFs for modelling the blood pool and myocardial surfaces lead to the concern of having to sample pseudo ground-truth distances outside the surface for supervision. Having read through Kuipers et al. (MIDL 2025) and Gropp et al. (2020), we find the approach elegant and interesting, and wish we would have been aware of this line of research earlier.
> > We must admit, we also like the implications SDF have on the topological hierarchies that can be formulated (just like the reviewer stated regarding the LV myocardium class being able to be modelled as a superset of the LV blood-pool). We will be experimenting with this in future versions of our projects.
> >
> > We are also curious about whether the reviewer has any honest remarks on potential pitfalls of using SDFs over occupancies that might not be immediately obvious at first glance, or whether SDFs are an all-around improvement over occupancies.
> > One initial concern we had about Eikonal-constrained SDFs regarded how they interact with extreme anisotropicity/sparsity. In CMR data, short axis slices have a 10mm through-plane resolution (as opposed to the <2mm in plane resolution). This is unlike the data used in Kuipers et al. (MIDL 2025). We are not able think of a reason why an INR could not be able to find a plausible smooth solution to the SDF in apical/basal regions where surfaces run perpendicular to the low-resolution dimension, but perhaps the reviewer can offer pitfalls of SDFs (if any, whether theoretical or practical) that are not readily obvious from the literature.

---

> > > ### Comment · Reviewer_1PVa · 2026-01-29
> > > **Remarks on SDF learning**
> > >
> > > I appreciate the authors' engagement in the Discussion, and am happy to hear it might inspire future directions. From both personal experience and a topological point of view, I would argue that SDFs, when trained properly as described in the mentioned references, are an overall upgrade to occupancy fields. The additional inductive bias explicitly enforces the INR to construct shapes rather than perform coordinate-wise classification. Additionally, SDFs can easily be converted into meshes by sampling their zero-level set, which can be done with existing Python implementations of marching cubes.
> > >
> > > Regarding resolution, this might indeed be a constraint for individually trained INRs. However, given a population-based approach, as proposed here, I believe an SDF function should be able to generalize as effectively. Optionally, shapes can be learned in different modalities with higher resolution, and subsequently be applied to the target modality (for which authors could draw inspiration from [1]).
> > >
> > > I would appreciate such a Discussion on future work to find its place in the camera-ready version, but this is up to the authors. Nevertheless, an active discussion such as we have engaged in here is, in my opinion, what merits the research community as a whole, and for that I will raise my score to a strong accept.
> > > I look forward to seeing future work from the authors!
> > >
> > > [1] Sander, J., de Vos, B. D., Bruns, S., Planken, N., Viergever, M. A., Leiner, T., & Išgum, I. (2023). Reconstruction and completion of high-resolution 3D cardiac shapes using anisotropic CMRI segmentations and continuous implicit neural representations. Computers in Biology and Medicine, 164, 107266.

---

### Official Review · Reviewer_LaTh · 2026-01-10

**Confidence:** 4
**Preliminary Rating:** 4

**Summary:**

is paper presents an approach to derive a neural implicit representation for 3D+t cardiac functions, in the form of voxel intensity and segmentation probability, from 2D MRI views. This function is also conditioned on a subject-specific conditioning vector that is optimized from the data. The method also simultaneously optimizes rotation and translation parameters per slice, as well as a per-slice intensity scaling matrix. The method was experimented on 120 subjects from the UK-biobank.

**Strengths:**

The method is overall interesting, creating an ability to optimize a 3d+t function from partial 2D views.

The consideration of motion correction and physics-based intensity modeling are interesting.

The initial experimental results demonstrated the effect of input images and motion correction on the generated results.

**Weaknesses:**

It is not clear whether the training loss in Section 3.3 is optimized per subject, or across all subjects.

The testing procedure on “new” subjects is not clear: whether re-optimization is needed (see details below).

There was no consideration of comparison to relevant works.

**Detailed Comments:**

Since there are per patient embeddings, motion parameters, and image intensity scaling parameters being optimized simultaneously with the INR, it is assumed that the INR needs to be trained per patient. Two questions related to this aspects are not clear and can be better clarified.

1. Is the optimization as described carried out per patient, or across N patients? i.e., is the INR parameters shared across training patients? The “7-day” training reported is for training per subject, or all training subjects?

2. For a new subject, at inference time, wouldn’t the optimization process need to be carried out again because of the need of the patient-specific h embedding along with the motion/intensity scaling matrices? Appendix B included results from “inference on new subjects”, leaving the impression that the trained INR can be directly applied without optimization — but what about the patient-specific parameters that need to be optimized as well?

**Justification Of The Preliminary Rating:**

This is a proof-of-concept of a novel method underpinned by several interesting ideas. While comparison to baselines is limited, it may be okay given the nature of the initial works. There are some confusion that need to be clarified, but should be addressable through the rebuttal.

**Questions To Address In The Rebuttal:**

1. Questions about the training and test procedures as detailed above are the most important to clarify for rebuttal

2. Some comparison to related works, or justifications for why such comparison is not feasible/needed, will help.

---

> ### Author Response · Authors · 2026-01-24
> **Clarification on optimization process and feasibility of a quantitative analysis**
>
> We are happy to see the reviewer shares our interest in the novelty of the approach. We structure the rebuttal in two sections: **clarifications over the optimization process**, and **discussion over the feasibility of a quantitative analysis**.
>
> ## Clarifications over the optimization process
>
> > Is the optimization as described carried out per patient, or across N patients? i.e., is the INR parameters shared across training patients?
>
> The training process is carried out over all N subjects using the same network parameters $\psi$. This is standard practice in auto-regressive approaches as popularized by DeepSDF (Park et. al. 2019), where subject-specific deviations are modeled via the given subject’s conditioning vector $\mathbf{h}$. This conditioning vector is typically fed to the shared network via the input (although alternative methods exist). While we describe this in the *network architecture* section, following the comments offered by the reviewer, we do see a need to add explicit clarification for readers who may not be fully familiar with this style of architecture. We have since added clarification in that same section:
> *... While the same network weights $\psi$ are shared across all subjects, subject-specific characteristics are modeled using a given subject's learnable conditioning vector $\mathbf{h}$.*
> Additionally, we have added further clarification in the loss function equation in section 3.3:
> *... The network parameters $\psi$ are shared across all voxels, independent of which slice or subject a voxel originates from. …*
> We want to further note that if the INR parameters $\psi$ were not shared, there would be no need for a latent vector. From early tests in our research, subject specific networks (no parameter sharing across subjects) perform very poorly due to the sparsity of the data. Generalization to new subject for the segmentation task would also not be possible (unless using wildly different approaches such as meta-learning (Vyas et. al. 2025)).
>
>
> > The “7-day” training reported is for training per subject, or all training subjects? For a new subject, at inference time, wouldn’t the optimization process need to be carried out again because of the need of the patient-specific h embedding along with the motion/intensity scaling matrices?
>
> The 7 day training process is performed across all subjects, during which the network with parameters $\psi$ is supervised using  randomly sampled voxels originating from all of these subjects.
> As described in the **Inference** section of section 3.3, the inference process of this type of architecture involves optimization of subject-specific parameters. This is standard methodology in DeepSDF-style of architectures, wherein a new subject-specific latent vector is regressed from the available observations (voxels in our case) via back-propagation through (the frozen weights of) the network. This is obviously not the same time of inference one expects from traditional grid-based networks (such as a UNet). Regressing this latent does take a significant amount of time (2 minutes per subject in our case), and is a known downside of this type of modeling which has led to a wide range of ongoing literature suggesting improvements and work-arounds.
>
> This inference process can already be found in the original version of our paper in section 3.3:
>
> **Inference:** *To infer ... a new subject, a new latent vector, rotation, translation, and intensity scaling parameters are initialized as zeros, and are optimized exclusively on available image information ... . These parameters are optimized by backpropagating gradients through $f_\mathcal{I}$ while keeping weights $\psi$ frozen for a total of $2500$ steps (2 minutes)...*
>
>
>
> ## Discussion over the feasibility of a quantitative analysis
> > Some comparison to related works, or justifications for why such comparison is not feasible/needed.
>
> As this is a concern raised by multiple reviewers, we have included a general discussed in the shared rebuttal section. We refer the reviewer to that rebuttal section for a deeper discussion. In short, our work serves a shape-modelling task which is a fundamentally different task than in-plane segmentation networks perform. While in-plane segmentation performance is somewhat of a proxy to shape modelling accuracy, CMR data and its annotation process has many flaws: ground-truth alignment is not known, openly available segmentation labels are often flawed, and labels are impossible to be made consistent across views by performing slice-wise annotation. Optimizing for in-plane segmentation as grid-based segmentation models (such as a UNet) currently do, would be a different task than modeling plausible continuous shapes. We fear this may mislead readers into thinking our work falls into the same in-plane paradigm and underperforms at it, while in fact the nature of modeling view-consistent 3D shapes guarantees it is impossible to ever achieve perfect in-plane metrics.

---

> > ### Comment · Reviewer_LaTh · 2026-02-02
> >
> > I'd like to thank the authors for the clarifications. It is important to make sure that the final manuscript include clear information about the need to optimize per-subject embeddings at both training and test time. The absence of comparative works to baselines is acceptable for a proof-of-concept. I will stay with my original rating.

---

### Official Review · Reviewer_krwG · 2026-01-10

**Confidence:** 4
**Preliminary Rating:** 3
**Final Rating:** 4

**Summary:**

Cardiac MRI (CMR) acquires heart images on cross-sectional planes, which requires aligning the motion-distorted two dimensional slice images on the heart anatomy.
The study presents a supervised method utilizing segmentation labels, that tolerate motion during the joint reconstruction and segmentation of new samples.
They apply an implicit neural network on three dimensional point clouds formed by CMR slice-data, whose segmentation labels are produced synthetically in Bai et. al.
The suggested method NISF++ extends NISF architecture.
The rigid transformation contrains the search space to align samples on a canonical anatomy, further supported with a physics informed modeling.
 The reconstruction performance at each slice order and inter-slice intensity differences, segmentation differences are evaluated, PSNR, SSIM, DICE, NCC metrics.
The results show on-par reconstruction performance with Bai et. al. Motion correction improves intensity and segmentation difference on slice planes.

**Strengths:**

The motion correction mechanism simplifies the anatomical alignment problem across samples.

The integration of physics-informed modeling is well-motivated.

The paper is generally well-structured and clearly written.

The visual ablation using the point spread function effectively demonstrates the method's capability in correcting imaging artifacts, providing clear empirical evidence.

**Weaknesses:**

As with many implicit neural representations, the method inherently produces a spatially smoothed or averaged reconstruction. This risks blurring fine anatomical details. The motion correction mechanism, while beneficial for alignment, may exacerbate this blurring effect, potentially degrading in-plane resolution.

The study does not involve an intuitive baseline.

The loss function combines multiple terms, but their individual contributions are not analyzed. An ablation study quantifying the impact of each loss function term is necessary to understand what drives performance.

Validation by visual inspection is hard to carry out exhaustively and the finding does not rule out that NISF++ may also lead a sample to a worse reconstruction. See questions section.

The suggested method is trained on an NVIDIA A100 GPU, lasting a total of 7 days, which is a computationally demanding approach.

Minor: Suggested method is a solid improvement but not a foundational approach.

Typo: figure 1 caption "ridid-body" -> "rigid-body "

**Detailed Comments:**

This method employs synthetically generated labels that contain errors. This presents a dual effect; it enables the demonstrated gains but also fundamentally caps accuracy and introduces error propagation. Consequently, the assertion of robustness is not fully substantiated, resting on a limited empirical basis.

The results are compared to Bai et al., but the corresponding performance values from that study are omitted from the table. Including these reference values is necessary for a transparent comparison.

**Justification Of Final Rating:**

The authors have been highly responsive. While some comparative and quantitative gaps remain, the exploratory visual alignment evidence, clarified broader impact and clearer methodology meaningfully improve the manuscript. On balance, these revisions justify raising my score to weak accept.

**Justification Of The Preliminary Rating:**

The paper proposes a technically sound and novel integration of implicit representations, motion correction, and physics modeling. However, its current evaluation is insufficient to support its claims fully. The primary shortcomings are the lack of ablation studies and the absence of comparisons to standard or alternative methods.
While the paper critiques the quality of synthetic labels from prior work, it does not demonstrate that NISF++ convincingly outperforms other potential approaches that could use the same data. Addressing these comparative and analytical gaps is crucial for validation.

**Questions To Address In The Rebuttal:**

The assertion of improved slice alignment would be strengthened by a more systematic visual analysis. Providing a broader set of visual results would help readers transparently assess this claim.

---

> ### Author Response · Authors · 2026-01-25
> **Visual analysis, baselines, CMR data, and compute (part 1)**
>
> We thank the reviewer for the comments and the detailed criticism of our manuscript.
>
> ## Visual analysis of slice alignment
>
> > The assertion of improved slice alignment would be strengthened by a more systematic visual analysis. Providing a broader set of visual results would help readers transparently assess this claim.
>
> Following the reviewer’s criticism, we have added a new figure (Figure 4) displaying pre- and post-optimization intensities samples along slice intersections. These results more rigorously portray how prior to optimization, intensities may not spatially match across views. However, after optimization, intensities along intersection lines (vertical sections in the figure) can be seen to display spatially matching landmarks.
>
>
> ## High compute costs:
>
> > The suggested method is trained on an NVIDIA A100 GPU, lasting a total of 7 days, which is a computationally demanding approach.
>
> We refer the reviewer to the general rebuttal submission for a deeper discussion.
> Implicit neural representation are notoriously slow to train in multi-subject settings. Our approach has similar training times to adjacent works. We have extended the discussion section in the manuscript with a paragraph discussing this issue, as well as mentioning ongoing research avenues that optimize training time.
>
> ## Blurring effect in INRs
>
> > As with many implicit neural representations, the method inherently produces a spatially smoothed or averaged reconstruction. This risks blurring fine anatomical details. The motion correction mechanism, while beneficial for alignment, may exacerbate this blurring effect, potentially degrading in-plane resolution.
>
> While much work has been put towards improving implicit neural representations’ modeling fidelity towards high-frequency features, multi-subject architectures are notoriously known to produce blurry reconstructions during inference time.
> The leading theory is that during optimization of the training set, the network only learns to reconstruct the sub-regions of the latent space where training latents are available. When performing inference by ‘finding’ a new latent vector given an unseen subject’s data, the model converges to areas of the latent space which it has never seen before and thus can’t precisely model low-level details.
>
> However, since our work prioritizes the shape modeling component, which is a low-frequency function, we are not too bothered by the reconstruction blurriness pointed out by the reviewer. It nonetheless something we would like to overcome and are working on an extension of the approach which investigates solutions to this low-frequency bias by smarter sampling and training using latent distributions.

---

> > ### Author Response · Authors · 2026-01-25
> > **(part 2)**
> >
> > ## Lack of baselines and limitations of data
> >
> >
> > We want to begin by listing what we see as the current limitations in the field of cardiac modelling that led us to motivate our approach.
> >
> > ### The chicken/egg problem with slice-alignment and 3D modelling:
> >
> > The misalignment of slices in CMR data is a well-known issue even in curated datasets such as the UK BioBank. The vast majority of motion-correction works make use of explicit statistical shapes, whereby slice-wise contours are used to ‘guide’ the slice to the 3D statistical shape. However, these 3D shapes have to be extracted from what is assumed to be motion-free cohorts or from subject-specific complementary scans in other imaging modalities. The handful of works that stray away from using shape information require slice intersections to be present and perform slice alignment naively without physics constraints in intensity measurements.
> >
> > To the best of our knowledge, we are the first to formulate a solution to this problem using a joint modelling-alignment task. *We respectfully disagree with the reviewer’s claim of our work being simply “a solid improvement but not a foundational approach”*. We believe our work to be in line with the contributions that NeSVoR (Xu et. al. 2023) [1] had to fetal MR imaging (where every previous existing work was performing super-resolution and motion correction as separate tasks, having to rely on strong heuristics and largely unable to be learned end-to-end), or “Segment Anything in 3D with NeRFs” [2] (where 2D SAM models are employed to generate 3D shapes on data that would otherwise be prohibitively expensive to annotate).
> >
> >
> > This is also the reason behind not being able to include (meaningful) baselines for our work. We refer the reviewer to the general rebuttal submission for a deeper discussion.
> >
> > In short, in-plane segmentation models perform a fundamentally different task than the object modeling task our approach undertakes. While in-plane segmentation performance is somewhat of a proxy to shape modelling accuracy, CMR data and its annotation process has many flaws: ground-truth alignment is not known, openly available segmentation labels are often flawed, and labels are impossible to be made consistent across views by performing slice-wise annotation. Optimizing for in-plane segmentation as grid-based segmentation models (such as a UNet) currently do, would be a different task than modeling slice-consistent plausible 3D shapes. We fear this may mislead readers into thinking our work falls into the same in-plane paradigm and underperforms at it, while in fact the nature of modeling view-consistent 3D shapes guarantees it is impossible to ever achieve perfect in-plane metrics.
> >
> > Existing segmentation works can only produce segmentations in-plane and are unable to integrate multiple views. Having run nn-UNet baselines in the past, our experience is that these in-plane models overfit to the general flaws of in-plane segmentations labels, especially for the apical and basal regions (see next section). We find that at the current levels of performance, the in-plane metrics of traditional segmentation networks stop offering a meaningful quantification of shape correctness. In the worst cases, these metrics may mislead readers by portraying better performance, when in fact they suffer from the issues outlined in the paper.
> > While this is obvious for the limited accuracy of the automatic labels used in our work (see next section), human annotations also suffer from them due to being created slice-wise.
> >
> > ### Lack of datasets and segmentation tools:
> >
> > The field of cardiac modeling suffers from an absence of open-source resources. Public segmentation models are poor, especially in the apical and basal regions. The original data to train these models is locked behind privacy constraints. In the few cases where datasets are made available, there are no complementary short-axis and long-axis imaging modalities. The short-axis model trained by Bai. et. al. continues to be the best available resource despite being a UNet trained in 2019. A similar issue is present for works that employ super-resolution meshes, the meshes (nor their source data) are often not made available, and in the few cases when they are, they are constructed using exclusively short-axis images. We thus used these synthetic ground-truth due to having no other large-scale alternative.
> >
> >
> > ### Citations
> >
> > [1] Xu J, Moyer D, Gagoski B, Iglesias JE, Grant PE, Golland P, Adalsteinsson E. NeSVoR: implicit neural representation for slice-to-volume reconstruction in MRI. IEEE transactions on medical imaging. 2023 Jan 11;42(6):1707-19.
> >
> > [2] Cen J, Zhou Z, Fang J, Shen W, Xie L, Jiang D, Zhang X, Tian Q. Segment anything in 3d with nerfs. Advances in Neural Information Processing Systems. 2023 Dec 15;36:25971-90.

---

### Author Response · Authors · 2026-01-24
**General rebuttal on novelty, baselines, data limitations, and compute resources (part 1)**

Dear Reviewers, dear ACs,

We sincerely appreciate the time and effort you have dedicated to assessing our manuscript. We are excited and encouraged by the comments characterizing our work as a “novel integration of implicit representations” (R1) using “several interesting ideas” (R2), leading to a “clinically relevant problem formulation” and thus to an “elegant unified framework” (R3).

In response to the reviewers' questions and their valuable feedback, we have updated and carefully revised the manuscript. We briefly provide clarification regarding commonly raised points here, and refer to our individual responses for detailed answers to specific questions.

## Novelty, contextualization within related work, applicability of baselines,  and data limitations

Our work is motivated by what we see as a general limitation of in-plane segmentation tasks, where in-plane accuracy is considered a sufficient proxy for shape correctness. For many clinical applications, acquiring rough volumetric or tissue strain measurements via in-plane segmentation has been shown to be an effective approach. The thesis of our paper argues that, regarding the task of modeling the cardiac muscle, the task of traditional in-plane segmentation can only be taken so far. As such, our approach proposes using CMR data under the lens of an object modeling task. We describe the limitations of traditional CMR segmentation baselines as two-fold: (1) issues regarding the nature of cardiac MR data, and (2) issues involving the in-plane architectures of the established image segmentation paradigm.

(1) - The acquisition process in cardiac MR makes it impossible to guarantee no motion is present between intensities observed across slices. On top of that, the slice-wise annotation process (whether automatic or performed by a human) also can’t guarantee consistency across slices. This is especially the case for the few available open-source datasets. This results in a modeling task where no true ground-truth is available for a full cardiac shape. Furthermore, highly-curated datasets that would make such tasks possible are beyond anything that is currently available outside highly specialized MR research modalities and prohibitively expensive to annotate. For these reasons (and quite reasonably so), segmentation tasks have predominantly remained an in-plane task for grid-based deep learning models, leaving any form of shape modeling as a niche task to be performed on specialized high-resolution scans outside standardized clinical settings.
To the best of our knowledge, we are the first to offer a solution to this problem using a joint modelling-alignment task. Nonetheless, our approach is *purely exploratory*. Not only are there no existing works aiming to model cardiac shapes directly from arbitrarily oriented imaging slices, there is no ground-truth to quantify the performance of our out-of-plane predictions.
We believe our work to be in line with the contributions that NeSVoR (Xu et. al. 2023) [1] had to fetal MR imaging (where every previous existing work was performing super-resolution and motion correction as separate tasks, having to rely on strong heuristics and largely unable to be learned end-to-end), or “Segment Anything in 3D with NeRFs” [2] (where 2D SAM models are employed to generate 3D shapes on data that would otherwise be prohibitively expensive to annotate).

(2) - Our reasoning behind not providing in-plane baselines is that we do not believe our approach is intended as a replacement of those methods, as the task we solve is inherently different for the reasons outlined. We fear it may mislead readers by fixating on proxy metrics, as opposed to considering the limitations of the in-plane segmentation paradigm. We use in-place Dice scores as an indirect proxy of performance to give readers an intuition on modeling accuracy and alignment improvement, but shape assessment remains a largely qualitative exercise. In fact, **the nature of modeling view-consistent 3D shapes guarantees it is impossible to ever achieve perfect in-plane metrics**. Having run nn-UNet baselines in the past, our experience is that these in-plane models overfit to the general flaws of in-plane segmentations labels, especially for the apical and basal regions. We find that at the current levels of performance, the in-plane metrics of traditional segmentation networks stop offering a meaningful quantification of shape correctness.

---

> ### Author Response · Authors · 2026-01-24
> **(part 2)**
>
> ## Training Resources:
> We appreciate the comments on computational overhead. While our proposed method requires a 7-day training period on a single GPU, this is a one-time cost for the entire UK Biobank cohort. In stark contrast to other INR-based approaches that require computationally expensive optimization for every single subject, our approach is trained once and generalizes across the population.
> While the field of deep learning generally calls for long training times, similar implicit representation works report similar values: 9 days for NISF [4], 4 days for NIMOSEF [5]. While these training times obviously depend on cohort size, they are also generally dependent on function complexity. Low-frequency functions such as shapes are generally cheaper to train (12 hours for shape modeling tasks [6]), while high-detail representations such as in the original NeRF [3] (which is not cohort-based) are reported to take 1-2 days per scene. Nonetheless, we consider the resource requirements to not be far off from standard ranges for high-performance models (e.g., nnU-Net).
>
> We have included a paragraph in the discussion section of the manuscript commenting on this topic. We also mention on-going works that tackle optimization efficiency (eg. via sampling strategies, or simpler primitives beyond neural networks).
>
>
>
> ### Citations
>
> [1] Xu J, Moyer D, Gagoski B, Iglesias JE, Grant PE, Golland P, Adalsteinsson E. NeSVoR: implicit neural representation for slice-to-volume reconstruction in MRI. IEEE transactions on medical imaging. 2023 Jan 11;42(6):1707-19.
>
> [2] Cen J, Zhou Z, Fang J, Shen W, Xie L, Jiang D, Zhang X, Tian Q. Segment anything in 3d with nerfs. Advances in Neural Information Processing Systems. 2023 Dec 15;36:25971-90.
>
> [3] Mildenhall B, Srinivasan PP, Tancik M, Barron JT, Ramamoorthi R, Ng R. Nerf: Representing scenes as neural radiance fields for view synthesis. Communications of the ACM. 2021 Dec 17;65(1):99-106.
>
> [4] Stolt-Ansó N, McGinnis J, Pan J, Hammernik K, Rueckert D. Nisf: Neural implicit segmentation functions. InInternational Conference on Medical Image Computing and Computer-Assisted Intervention 2023 Oct 1 (pp. 734-744). Cham: Springer Nature Switzerland.
>
> [5] Banus J, Delaloye A, M. Gordaliza P, Georgantas C, van Heeswijk RB, Richiardi J. NIMOSEF: Neural Implicit Motion and Segmentation Functions. InInternational Conference on Medical Image Computing and Computer-Assisted Intervention 2025 Sep 20 (pp. 444-454). Cham: Springer Nature Switzerland.
>
> [6] Amiranashvili T, Lüdke D, Li HB, Menze B, Zachow S. Learning shape reconstruction from sparse measurements with neural implicit functions. InInternational Conference on Medical Imaging with Deep Learning 2022 Dec 4 (pp. 22-34). PMLR.

---

### Author Rebuttal · Authors · 2026-01-25

**Rebuttal:**

As listed in the official comment, as well as in individual reviewer responses, we have added:
- Additional qualitative figure for intensity alignment before and after optimization.
- Added clarification on parameter sharing across all subject.
-  Added discussion section regarding computation costs.

These changes have been highlighted in the manuscript.

**Supporting Material:**

/attachment/26024e9d617b692c6ace6511abea87070600d20d.pdf

---

### Comment · Area_Chair_z3Pq · 2026-01-29
**Official Comment by Area Chair**

Dear Reviewers:

We kindly encourage you to take a moment to review the authors’ rebuttals and submit your feedback. Your prompt feedback is important for ensuring a thorough review. Thank you for your contributions to MIDL 2026. If you have responded to the authors' rebuttal, please feel free to ignore this message.

Thanks, AC

---

### Comment · Area_Chair_z3Pq · 2026-02-01
**Official Comment by Area Chair**

Dear Reviewers:

Please take a moment to update their final rating by clicking “Edit” → “Official Review” and providing the Final Rating by February 1st 2026 (23:59 AoE).

Thanks, AC

---

### Meta-Review · Area_Chair_z3Pq · 2026-02-07

**Recommendation:** Accept (Poster)
**Confidence:** 5

**Metareview:**

All reviewers found the proposed method to be novel and the results promising. The AC thus recommended acceptance of the paper.

---

### Decision · Program_Chairs · 2026-02-13

Accept (Poster)